# Algebraic tests of general Gaussian latent tree models

**Dennis Leung**
Department of Data Sciences and Operations
University of Southern California
dmhleung@uw.edu

**Mathias Drton**
Department of Statistics, University of Washington &
Department of Mathematical Sciences, University of Copenhagen
md5@uw.edu

## Abstract

We consider general Gaussian latent tree models in which the observed variables are not restricted to be leaves of the tree. Extending related recent work, we give a full semi-algebraic description of the set of covariance matrices of any such model. In other words, we find polynomial constraints that characterize when a matrix is the covariance matrix of a distribution in a given latent tree model. However, leveraging these constraints to test a given such model is often complicated by the number of constraints being large and by singularities of individual polynomials, which may invalidate standard approximations to relevant probability distributions. Illustrating with the star tree, we propose a new testing methodology that circumvents singularity issues by trading off some statistical estimation efficiency and handles cases with many constraints through recent advances on Gaussian approximation for maxima of sums of high-dimensional random vectors. Our test avoids the need to maximize the possibly multimodal likelihood function of such models and is applicable to models with larger number of variables. These points are illustrated in numerical experiments.

## 1 Introduction

Latent tree models are associated to a tree-structured graph in which some nodes represent observed variables and others represent unobserved (latent) variables. Due to their tractability, these models have found many applications in fields ranging from the traditional life sciences, biology and psychology to contemporary areas such as artificial intelligence and computer vision; refer to Mourad et al. [2013] for a comprehensive review. In this paper, we study the problem of testing the goodness-of-fit of a postulated Gaussian latent tree model to an observed dataset. In a low dimensional setting where the number of observed variables is small relative to the sample size at hand, testing is usually based on the likelihood ratio which measures the divergence in maximum likelihood between the postulated latent tree model and an unconstrained Gaussian model. This, however, requires maximization of the possibly multimodal likelihood function of latent tree models. In contrast, recent work of Shiers et al. [2016] takes a different approach and leverages known polynomial constraints on the covariance matrix of the observed variables in a given Gaussian latent tree. Specifically, the postulated latent tree is tested with an aggregate statistic formed from estimates of the polynomial quantities involved. This approach can be traced back to Spearman [1904] and Wishart [1928]; also see Drton et al. [2007, 2008].

We make the following new contributions. In Section 2, we extend the polynomial characterization of Shiers et al. [2016] to cases where observed nodes may also be inner nodes of the tree as considered,

for example, in the tree learning algorithms of Choi et al. [2011]. Section 3 describes how we may use polynomial equality constraints to test a star tree model. We base ourselves on the recent groundbreaking work of Chernozhukov et al. [2013a], form our test statistic as the maximum of unbiased estimates of the relevant polynomials, and calibrate the critical value for testing based on multiplier bootstrapping techniques. This new way of using the polynomials to furnish a test allows us to handle latent trees with a larger number of observed variables and avoids potential singularity issues caused by individual polynomials. Numerical experiments in Section 4 makes comparisons to the likelihood ratio test and assesses the size of our tests in finite samples. Section 5 discusses future research directions.

**Notation.** Let $1 \leq r \leq m$ be two positive integers. We let $[m] = \{1, \ldots, m\}$ and write $\left\{ {m \atop r} \right\} := \{I \subseteq [m] : |I| = r\}$ for the collection of subsets of $[m]$ with cardinality $r$. The supremum norm of a vector is written $\| \cdot \|_\infty$. For two random variables $R_1$ and $R_2$, the symbols $R_1 =_d R_2$ indicates that $R_1$ and $R_2$ have the same distributions, and $R_1 \approx_d R_2$ indicates that the distributions are approximately equal. $N(\mu, \sigma^2)$ means a normal distribution with mean $\mu$ and standard deviation $\sigma^2$.

## 2 Characterization of general Gaussian latent trees

We first provide the definition of the models considered in this paper. A tree is an undirected graph in which any two nodes are connected by precisely one path. Let $T = (V, E)$ be a tree, where $V$ is the set of nodes, and $E$ is the set of edges which we take to be unordered duples of nodes in $V$. We say that $T$ is a latent tree if it is paired with a set $\mathbf{X} = \{X_1, \ldots, X_m\} \subset V$, corresponding to $m$ observed variables, such that $v \in \mathbf{X}$ whenever $v \in V$ is of degree less than or equal to two. In particular, $\mathbf{X}$ contains all leaf nodes of the tree $T$ (i.e., nodes of degree 1), but it may contain additional nodes. The nodes in $V \setminus \mathbf{X}$ correspond to latent variables that are not observed but each have at least three other neighbors in the tree. This minimal degree requirements of 3 on the latent nodes ensures identifiability [Choi et al., 2011, p.1778]. In the terminology of mathematical phylogenetics, $T$ is a semi-labeled tree on $\mathbf{X}$ with an injective labeling map; see Semple and Steel [2003, p.16]. However, *phylogenetic trees* are latent trees restricted to have $\mathbf{X}$ equal to the set of leaves. While we have defined $\mathbf{X}$ as a set of nodes, it will be convenient to abuse notation slightly and let $\mathbf{X}$ also denote a random vector $(X_1, \ldots, X_m)'$ whose coordinates correspond to the nodes in question. The context will clarify whether we refer to nodes or random variables.

Now we present the polynomial characterization of a Gaussian latent tree graphical model that extends the results in Shiers et al. [2016]. The Gaussian graphical model on $T$, denoted $\mathcal{M}(T)$, is the set of all $|V|$-variate Gaussian distributions respecting the pairwise Markov property of $T$, i.e., for any pair $u, v \in V$ with $(u, v) \notin E$, the random variables associated to $u$ and $v$ are conditionally independent given the variables corresponding to $V \setminus \{u, v\}$. The $T$-Gaussian latent tree model on $\mathbf{X}$, denoted $\mathcal{M}_\mathbf{X}(T)$, is the set of all $m$-variate Gaussian distributions that are the marginal distribution for $\mathbf{X}$ under some distribution in $\mathcal{M}(T)$. For a given distribution in $\mathcal{M}(T)$, let $\rho_{pq}$ be the Pearson correlation of the pair $(X_p, X_q)$ for any $1 \leq p \neq q \leq m$. The pairwise Markov property implies that

$$\rho_{pq} = \prod_{(u,v) \in ph_T(X_p, X_q)} \rho'_{uv}, \tag{2.1}$$

where $ph_T(X_p, X_q)$ denotes the set of edges on the unique path that connects $X_p$ and $X_q$ in $T$, and $\rho'_{uv}$ is the Pearson correlation between a pair of nodes $u$ and $v$ in $V$. Of course, $\rho'_{uv} = \rho_{pq}$ if $u = X_p$ and $v = X_q$. In the sequel, we often abbreviate $ph_T(X_p, X_q)$ as $ph_T(p, q)$ for simplicity.

Suppose $\Sigma = (\sigma_{pq})_{1 \leq p,q \leq m}$ is the covariance matrix of $\mathbf{X}$. Our task is to test whether $\Sigma$ comes from $\mathcal{M}_\mathbf{X}(T)$ against a saturated Gaussian graphical model. We assume that all edges in the tree $T$ correspond to a nonzero correlation, so that $\Sigma$ contains no zero entries. The covariance matrices for $\mathcal{M}_\mathbf{X}(T)$ are parametrized via (2.1). As shown in Shiers et al. [2016], this set of covariance matrices may be characterized by leveraging results on pseudo-metrics defined on $\mathbf{X}$. Suppose $w : E \longrightarrow \mathbb{R}_{\geq 0}$ is a function that assigns non-negative weights to the edges in $E$. One can then define a pseudo-metric $\delta_w : \mathbf{X} \times \mathbf{X} \longrightarrow \mathbb{R}_{\geq 0}$ by

$$\delta_w(X_p, X_q) = \left\{ \begin{array}{ll} \sum_{e \in ph_T(p,q)} w(e) & : p \neq q, \\ 0 & : p = q. \end{array} \right.$$

This is known as a $T$-induced pseudo-metric on $\mathbf{X}$. The following lemma characterizes all the pseudo-metrics on $\mathbf{X}$ that are $T$-induced. The proof is a bit delicate and is given in our supplementary material.

**Lemma 2.1.** *Suppose $\delta : \mathbf{X} \times \mathbf{X} \longrightarrow \mathbb{R}_{\geq 0}$ is a pseudo-metric defined on $\mathbf{X}$. Let $\delta_{pq} = \delta(X_p, X_q)$ for any $p, q \in [m]$ for simplicity. Then $\delta$ is a $T$-induced pseudo-metric if and only if for any four distinct $1 \leq p, q, r, s \leq m$ such that $ph_T(p,q) \cap ph_T(r,s) = \emptyset$,*

$$\delta_{pq} + \delta_{rs} \leq \delta_{pr} + \delta_{qs} = \delta_{ps} + \delta_{qr}, \tag{2.2}$$

*and for any three distinct $1 \leq p, q, r \leq m$,*

$$\delta_{pq} + \delta_{qr} = \delta_{pr} \tag{2.3}$$

*if $ph_T(p,r) = ph_T(p,q) \cup ph_T(q,r)$.*

Lemma 2.1 modifies Corollary 1 in Shiers et al. [2016] by requiring the extra equality constraints in (2.3) concerning three distinct variable indices. For any subset $S \subset \mathbf{X}$, let $T|S$ be the restriction of $T$ to $S$, that is, the minimal subtree of $T$ induced by the elements in $S$ with all the nodes of degree two not in $S$ suppressed [Semple and Steel, 2003, p.110]; refer to Section 7 in our supplementary material for the related graphical notions. Shiers et al. [2016] only consider phylogenetic trees in which the observed variables $\mathbf{X}$ always correspond to the set of nodes in $T$ with degree one. In this case the constraint in (2.3) is vacuous. Indeed, if $X_p, X_q, X_r$ are any three observed nodes in $T$, then $T|\{X_p, X_q, X_r\}$ must have the configuration on the left panel of Figure 2.1, and it can be seen that $ph_T(\pi_p, \pi_q) \cup ph_T(\pi_q, \pi_r) \neq ph_T(\pi_p, \pi_r)$ for any permutation $(\pi_p, \pi_q, \pi_r)$ of $(p, q, r)$. However, for a general latent tree $T$ whose observed nodes are not confined to be the leaves, condition (2.3) is necessary for a pseudo-metric $\delta$ to be $T$-induced: $T|\{X_p, X_q, X_r\}$ may take the configuration on the right panel of Figure 2.1, where for some permutation $(\pi_p, \pi_q, \pi_r)$ of $(p, q, r)$, $ph_T(\pi_p, \pi_r) = ph_T(\pi_p, \pi_q) \cup ph_T(\pi_q, \pi_r)$, and it must hold that

$$\delta_{\pi_p \pi_r} = \delta_{\pi_p \pi_q} + \delta_{\pi_q \pi_r}$$

if $\delta$ is $T$-induced.

While condition (2.2) appears in the result of Shiers et al. [2016], it may lead to different patterns of constraints for a general latent tree. For four distinct indices $1 \leq p, q, r, s \leq m$, there are three possible partitions into two subsets of equal sizes, namely, $\{p, q\}|\{r, s\}$, $\{p, r\}|\{q, s\}$ and $\{p, s\}|\{q, r\}$. These three partitions correspond to the path pairs

$$(ph_T(p,q), ph_T(r,s)), (ph_T(p,r), ph_T(q,s)) \text{ and } (ph_T(p,s), ph_T(q,r)) \tag{2.4}$$

respectively. Now refer to Figure 2.2 which shows all possible configurations of the restriction of $T$ to the four observed variables $X_p, X_q, X_r, X_s$. In Figure 2.2(a)-(c), up to permutations of the indices $\{p, q, r, s\}$, only one of three pairs in (2.4) can give an empty set when the intersection of its two component paths is taken. In light of (2.2), this implies that, for some permutation $\pi$ of the indices $p, q, r, s$,

$$\delta_{\pi_p \pi_q} + \delta_{\pi_r \pi_s} \leq \delta_{\pi_p \pi_r} + \delta_{\pi_q \pi_s} = \delta_{\pi_p \pi_s} + \delta_{\pi_q \pi_r}. \tag{2.5}$$

On the contrary, in Figure 2.2(d) and (e), it must be the case that each of the three path pairs in (2.4) gives an empty set when an intersection is taken between its two component paths, giving the equalities $\delta_{pq} + \delta_{rs} = \delta_{pr} + \delta_{qs} = \delta_{ps} + \delta_{qr}$ in consideration of (2.2).

Lemma 2.1 readily implies a characterization of the latent tree model $\mathcal{M}_{\mathbf{X}}(T)$ via polynomial constraints in the entries of the covariance matrix $\Sigma = (\sigma_{pq})$ as spelt out in the ensuing corollary. Its proof employs similar arguments in Shiers et al. [2016] and is deferred to our supplementary material. In what follows, we let $\mathcal{Q} \subset \binom{m}{4}$ be the set of all quadruples $\{p, q, r, s\} \in \binom{m}{4}$ such that only one of the three path pairs in (2.4) gives an empty set when the union of its two component paths is taken. In other words, $\mathcal{Q}$ contains all $S \in \binom{m}{4}$ such that $T|S$ is one of the configurations in Figure 2.2(a)-(c). Given $\{p, q, r, s\} \in \mathcal{Q}$, we write $\{p, q\}|\{r, s\} \in \mathcal{Q}$ to indicate that $\{p, q, r, s\}$ belongs to $\mathcal{Q}$ in a way that it is the path pairs $ph_T(p, q)$ and $ph_T(r, s)$ that have empty intersection. Similarly, we will let $\mathcal{L}$ be the set of all triples $S = \{p, q, r\} \in \binom{m}{3}$ such that $T|S$ has the configuration in Figure 2.1(b). We will use the notation $p - q - r \in \mathcal{L}$ to indicate that $q$ is the "middle point" such that $ph_T(p, q) \cap ph_T(q, r) = \emptyset$.

**Corollary 2.2.** *Suppose $\Sigma = (\sigma_{pq})_{1 \leq p, q \leq m}$ is the covariance matrix of $\mathbf{X}$ and has no zero entries. The following are together necessary and sufficient for the distribution of $\mathbf{X}$ to belong to $\mathcal{M}_{\mathbf{X}}(T)$:*

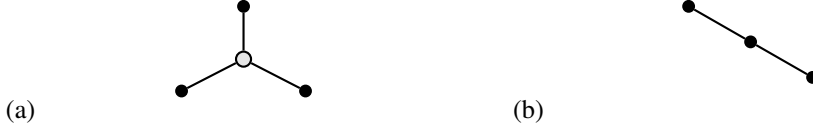

(a)                                                    (b)

Figure 2.1: The possible restrictions of a latent tree to three distinct observed variables. Observed variables correspond to solid black dots, latent variables to grey circles.

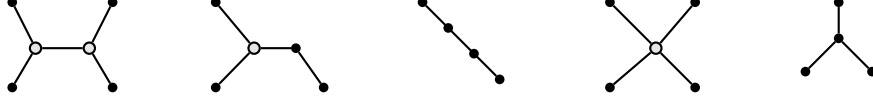

Figure 2.2: The possible restrictions of a latent tree to four distinct observed variables. From left to right, (a)-(e). Observed variables correspond to solid black dots, latent variables to grey circles.

   *i. Inequality constraints:*

      *(a) For any $\{p, q, r\} \in \binom{m}{3}$, $\sigma_{pq}\sigma_{pr}\sigma_{qr} \geq 0$.*

      *(b) For any $\{p, q, r\} \in \binom{m}{3} \backslash \mathcal{L}$,*

$$\sigma_{pq}^2\sigma_{qr}^2 - \sigma_{qq}^2\sigma_{pr}^2, \quad \sigma_{pr}^2\sigma_{qr}^2 - \sigma_{rr}^2\sigma_{pq}^2, \quad \sigma_{pq}^2\sigma_{pr}^2 - \sigma_{pp}^2\sigma_{qr}^2 \leq 0.$$

      *(c) For any $\{p, q\}|\{r, s\} \in \mathcal{Q}$, $\sigma_{pr}^2\sigma_{qs}^2 - \sigma_{pq}^2\sigma_{rs}^2 \leq 0$.*

   *ii. Equality constraints:*

      *(a) For any $p - q - r \in \mathcal{L}$, $\sigma_{pq}\sigma_{qr} - \sigma_{qq}\sigma_{pr} = 0$.*

      *(b) For any $\{p, q\}|\{r, s\} \in \mathcal{Q}$, $\sigma_{pr}\sigma_{qs} - \sigma_{ps}\sigma_{qr} = 0$.*

      *(c) For any $\{p, q, r, s\} \notin \mathcal{Q}$, $\sigma_{ps}\sigma_{qr} - \sigma_{pr}\sigma_{qs} = \sigma_{pq}\sigma_{rs} - \sigma_{pr}\sigma_{qs} = 0$.*

## 3   Testing a star tree model

In this section we illustrate how one can test a postulated Gaussian latent tree model using Corollary 2.2. In order to focus the discussion we treat the simple but important special case of a star tree, which corresponds to a single factor model. A single factor model with $m$ observed variables $\mathbf{X} = \{X_1 \ldots, X_m\}$ can be described by the linear system of equations

$$X_p = \mu_p + \beta_p H + \epsilon_p, \qquad 1 \leq p \leq m, \tag{3.1}$$

where $\mu_p$ is the mean of $X_p$, $H \sim N(0, 1)$ is a latent variable, $\beta_p$ is the loading coefficient for variable $X_p$, and $\epsilon_p \sim N(0, \sigma_{p,\epsilon}^2)$ is the idiosyncratic error for variable $X_p$. All of $H, \epsilon_1, \ldots, \epsilon_m$ are independent. The model postulates that $X_1, \ldots, X_m$ are conditionally independent given $H$. It thus corresponds to the graphical model associated with a star tree $T_\star = (V, E)$ with $V = \mathbf{X} \cup \{H\}$, $E = \{(H, X_p)\}_{1 \leq p \leq m}$.

Let $\mathbf{X}_1, \ldots, \mathbf{X}_n$ be i.i.d. draws from the distribution of $\mathbf{X}$, which is assumed to be Gaussian. Our goal is to test whether the distribution of $\mathbf{X}$ belongs to the single factor model $\mathcal{M}_{\mathbf{X}}(T_\star)$. Without loss of generality, we may assume that $\mu_p = 0$ for all $p \in [m]$ [Anderson, 2003, Theorem 3.3.2]. We proceed by testing whether all the constraints in Corollary 2.2 are simultaneously satisfied with respect to the latent tree $T_\star$. For simplicity, we will focus on testing the equality constraints in Corollary 2.2$(ii)$, and briefly discuss how one can incorporate the inequality constraints in Corollary 2.2$(i)$ in Section 5. For $T_\star$, both sets $\mathcal{L}$ and $\mathcal{Q}$ are empty, so that Corollary 2.2$(ii)(a)$ and $(b)$ are automatically satisfied. Hence, we are only left with Corollary 2.2$(ii)(c)$: For any $\{p, q, r, s\} \in \binom{m}{4}$,

$$\sigma_{ps}\sigma_{qr} - \sigma_{pr}\sigma_{qs} = \sigma_{pq}\sigma_{rs} - \sigma_{pr}\sigma_{qs} = 0. \tag{3.2}$$

The two polynomials above, equal to $\det(\Sigma_{pq,sr})$ and $\det(\Sigma_{ps,qr})$ respectively, are known as *tetrads* in the literature of factor analysis. It is well-known that they define equality constraints for a single factor model [Bekker and de Leeuw, 1987, Bollen and Ting, 1993, Drton et al., 2007].

## 3.1 Estimating tetrads

The idea now is to estimate each one of the $2 \cdot \binom{m}{4}$ tetrads in (3.2), and aggregate the estimates in a test statistic. From the sample covariance matrix $S = (s_{pq}) = n^{-1} \sum_{i=1}^{n} \mathbf{X}_i \mathbf{X}_i^T$, a straightforward sample tetrad estimate, say $s_{ps}s_{qr} - s_{pr}s_{qs}$, can be computed. If one define the vectors $\mathbf{t} = (s_{ps}, s_{qr}, s_{pr}, s_{qs})'$ and $\mathbf{t}_0 = (\sigma_{ps}, \sigma_{qr}, \sigma_{pr}, \sigma_{qs})'$, as well as the function $g(\mathbf{t}) = s_{ps}s_{qr} - s_{pr}s_{qs}$, by the delta method it is expected that $\sqrt{n}(g(\mathbf{t}) - g(\mathbf{t}_0)) \to N(0, \nabla g(\mathbf{t}_0)' V \nabla g(\mathbf{t}_0))$, where $V$ is the limiting covariance matrix of $\sqrt{n}(\mathbf{t} - \mathbf{t}_0)$ and $\nabla g(\mathbf{t}_0)$ is the gradient of $g(\cdot)$ evaluated at $\mathbf{t}_0$. However, the distribution of this sample tetrad becomes asymptotically degenerate at singularities, that is, when the gradient $\nabla g(\mathbf{t}_0)$ vanishes, which happens if the underlying true covariances are zero [Drton and Xiao, 2016]. Consequently, a standardized sample tetrad cannot be well approximated by a normal distribution if the underlying correlations are weak. More generally, even for stronger correlations, we found it difficult to reliably estimate the variance of all sample tetrads in larger-scale models.

We propose alternative estimators for which sampling variability can be estimated more easily. Due to the independence of samples, the tetrad $\det(\Sigma_{pq,sr}) = \sigma_{ps}\sigma_{qr} - \sigma_{pr}\sigma_{qs}$ can be estimated unbiasedly with the differences

$$Y_{i,(pq)(sr)} := X_{p,i}X_{s,i}X_{q,i+1}X_{r,i+1} - X_{p,i}X_{r,i}X_{q,i+1}X_{s,i+1}, \quad i = 1, \ldots, n-1, \tag{3.3}$$

where the subscripts in $Y_{i,(pq)(sr)}$ is indicative of the row and column indices for the submatrix $\Sigma_{pq,sr}$. These differences can then be averaged for an estimate of the tetrad. Similarly, one can form $Y_{i,(ps)(qr)}$ to estimate $\det(\Sigma_{ps,qr})$ in (3.2). If we arrange all the tetrads from $\{\det(\Sigma_{pq,sr}), \det(\Sigma_{ps,qr})\}_{\{p,q,r,s\} \in \{^m_4\}}$ into a $2\binom{m}{4}$-vector $\Theta$, and correspondingly arrange the estimates $\{Y_{i,(pq)(sr)}, Y_{i,(ps)(qr)}\}_{\{p,q,r,s\} \in \{^m_4\}}$ into a $2\binom{m}{4}$-vector $\mathbf{Y}_i$ for each $i$, then the central limit theorem for *1-dependent* sums ensures that for sufficiently large sample size $n$ we have the distributional approximation

$$\sqrt{n-1}(\bar{\mathbf{Y}} - \Theta) \approx_d N(0, \Upsilon), \tag{3.4}$$

where $\bar{\mathbf{Y}} = (n-1)^{-1} \sum_{i=1}^{n-1} \mathbf{Y}_i$ and $\Upsilon = \mathrm{Cov}[\mathbf{Y}_1, \mathbf{Y}_1] + 2\mathrm{Cov}[\mathbf{Y}_1, \mathbf{Y}_2]$. The latter limiting covariance matrix will not degenerate to a singular matrix even if the underlying covariance matrix for $\mathbf{X}$ has zeros at which some of the tetrads are singular (i.e. have zero gradient).

## 3.2 Bootstrap test

The fact from (3.4) could serve as the starting point for a test of model $\mathcal{M}(T_\star)$. However, the normal approximation quickly becomes of concern when moving beyond a small number of variables $m$. Indeed, the dimension of $\Theta$, $2\binom{m}{4}$, may well be close to the sample size $n$, or even larger. For instance, if $n = 250$, for a model with merely 8 observed variables the dimension of $\Theta$ is already $2\binom{8}{4} = 140$, more than half the sample size. A recent work of Zhang and Wu [2017], which follows up on the groundbreaking paper of Chernozhukov et al. [2013a] on Gaussian approximation for maxima of high dimensional independent sums, suggests that while the approximation in (3.4) may be dubious, by taking a supremum norm on both sides, the Gaussian approximation

$$\sqrt{n-1}\|(\bar{\mathbf{Y}} - \Theta)\|_\infty \approx_d \|\mathbf{Z}\|_\infty, \tag{3.5}$$

where $\mathbf{Z} =_d N(0, \Upsilon)$, can be valid even the dimension of $\Theta$ is large compared to $n$. In fact, the original work of Chernozhukov et al. [2013a] suggested that asymptotically, the dimension can be sub-exponential in the sample size for the Gaussian approximation to hold. In what follows, we will discuss implementation of and experiments with a vanishing tetrad test based on (3.5). While it is possible to adapt the supporting theory for the present application, the technical details are involved and beyond the scope of this conference paper.

Since $\bar{\mathbf{Y}}$ from (3.4) and (3.5) is an estimator of the vector of tetrads $\Theta$, it is natural to use $\|\bar{\mathbf{Y}}\|_\infty$ as the test statistic and reject the model $\mathcal{M}(T_\star)$ for large values of $\|\bar{\mathbf{Y}}\|_\infty$. The Gaussian approximation (3.5) suggests that when $\mathcal{M}(T_\star)$ is true, i.e. $\Theta = 0$, $\sqrt{n-1}\|\mathbf{Y}\|_\infty$ is distributed as $\|\mathbf{Z}\|_\infty$. Nevertheless, to calibrate critical values based on the distribution of $\|\mathbf{Z}\|_\infty$, one must estimate the unknown covariance matrix $\Upsilon$. Zhang and Wu [2017] suggested the batched mean estimator

$$\hat{\Upsilon} = \frac{1}{B\omega} \sum_{b=1}^{\omega} \left(\sum_{i \in L_b} (\mathbf{Y}_i - \bar{\mathbf{Y}})\right) \left(\sum_{i \in L_b} (\mathbf{Y}_i - \bar{\mathbf{Y}})\right)^T, \tag{3.6}$$

where for a batch size $B$ and $\omega := \lfloor (n-1)/B \rfloor$ one considers the non-overlapping sets of samples $L_b = \{1 + (b-1)B, \ldots, bB\}$, $b = 1, \ldots, \omega$. The "batching" aims to capture the dependence among the $\mathbf{Y}_i$'s, and has been widely studied in the time series literature [Bühlmann, 2002, Lahiri, 2003]. If model $\mathcal{M}(T_\star)$ is true, then (3.5) yields that

$$\mathcal{T} := \sqrt{n-1}\|\operatorname{diag}(\hat{\Upsilon})^{-1/2}\bar{\mathbf{Y}}\|_\infty \approx_d \|\operatorname{diag}(\hat{\Upsilon})^{-1/2}\tilde{\mathbf{Z}}\|_\infty,$$

where the right-hand side is to interpreted conditionally on $\hat{\Upsilon}$, with $\tilde{\mathbf{Z}} \sim N(0, \hat{\Upsilon})$ and $\operatorname{diag}(\hat{\Upsilon})$ comprising only the diagonal of $\hat{\Upsilon}$. More precisely, for a fixed test level $\alpha \in (0,1)$, if we define $q_{1-\alpha}$ to be the *conditional* $(1-\alpha)$-quantile of the distribution of $\|\operatorname{diag}(\hat{\Upsilon})^{-1/2}\tilde{\mathbf{Z}}\|_\infty$ given $\hat{\Upsilon}$, then

$$P(\mathcal{T} > q_{1-\alpha}) \approx \alpha, \tag{3.7}$$

according to Zhang and Wu [2017, Corollary 5.4]. We will use $\mathcal{T}$ as our test statistic for the model $\mathcal{M}(T_\star)$, and calibrate the critical value based on (3.7) by simulating the conditional quantile $q_{1-\alpha}$ from $\|\operatorname{diag}(\hat{\Upsilon})^{-1/2}\tilde{\mathbf{Z}}\|_\infty$ for fixed $\hat{\Upsilon}$.

### 3.3 Implementation

While our above presentation invoked the estimate $\hat{\Upsilon}$, which is a matrix with $O(m^8)$ entries, we may in fact bypass the problem of computing such a large covariance matrix for the tetrad estimates. To simulate the conditional quantile $q_{1-\alpha}$ in (3.7), let $e_1, \ldots, e_\omega$ be i.i.d. standard normal random variables, and consider the expression

$$\left\|\frac{\operatorname{diag}(\hat{\Upsilon})^{-1/2}}{\sqrt{B\omega}}\sum_{b=1}^\omega e_b\left(\sum_{i \in L_b}(\mathbf{Y}_i - \bar{\mathbf{Y}})\right)\right\|_\infty, \tag{3.8}$$

which has exactly the same distribution as $\|\operatorname{diag}(\hat{\Upsilon})^{-1/2}\tilde{\mathbf{Z}}\|_\infty$ conditioning on the data $\mathbf{X}_1, \ldots, \mathbf{X}_n$. We emphasize the $O(m^4)$ diagonal entries of $\hat{\Upsilon}$ are easily computed as variances in (3.6). In conclusion, we perform the following multiplier bootstrap procedure: (i) Generate many, say $E = 1000$, sets of $\{e_1, \ldots, e_\omega\}$, (ii) evaluate (3.8) for each of these $E$ sets, and (iii) take $q_{1-\alpha}$ to be the $1-\alpha$ quantile from the resulting $E$ numbers. Despite the bootstrap being a computationally intensive process, it is not hard to see that the evaluation of (3.8) for all $E$ sets of multipliers will involve $O(m^4 n E)$ operations, which even for moderate $m$ is far less than the $O(m^8)$ operations needed to obtain an entire covariance matrix for all tetrads.

*Remark.* It is instructive to make a comparison with the testing methodology in Shiers et al. [2016], where the focus was on lower-dimensional applications. Suppose $\tau : \Sigma \mapsto \Theta$ is the function that maps the covariance matrix $\Sigma$ into the vector $\Theta$ of tetrads in (3.2). To test the vanishing of the tetrads, Shiers et al. [2016] form plug-in estimates $\hat{\Theta} = \tau(S)$ for $\Theta$ with the sample covariance matrix $S = n^{-1}\sum_{i=1}^n \mathbf{X}_i\mathbf{X}_i^T$. Letting $\operatorname{Var}[\tau(S)]$ be the covariance matrix for the $2\binom{m}{4}$-vector $\tau(S)$, they form a Hotelling's $T^2$ type statistic as

$$n\tau(S)^T(\widehat{\operatorname{Var}}[\tau(S)])^{-1}\tau(S), \tag{3.9}$$

where $\widehat{\operatorname{Var}}[\tau(S)]$ is a consistent estimate for $\operatorname{Var}[\tau(S)]$; see also Drton et al. [2008]. For a test of model $\mathcal{M}(T_\star)$, this statistic is now compared to a chi-square distribution with $2\binom{m}{4}$ degrees of freedom. While this calibration is justified for sufficiently large sample size $n$ by a joint normal approximation analogous to (3.5), it can be problematic for large $m$. Even more pressing can be the computational disadvantage that one explicitly uses the entire matrix $\widehat{\operatorname{Var}}[\tau(S)]$ with its $O(m^8)$ entries.

## 4 Numerical experiments

We now report on some experiments with the bootstrap test based on the sup-norm of the estimated tetrads $\mathcal{T}$ proposed in Section 3. In the implementation we always use $E = 1000$ sets of normal multipliers to simulate the quantile $q_{1-\alpha}$ and work with batch size $B = 3$ in (3.8). We also benchmark our methodology against the likelihood ratio test for factor models implemented by the function `factanal` in the `base` library of R, which implements a likelihood ratio (LR) test with Bartlett correction for more accurate asymptotic approximation. The critical value of the LR test is calibrated with the chi-square distribution with $\binom{m-1}{2} - 1$ degrees of freedom [Drton et al., 2009, p.99].

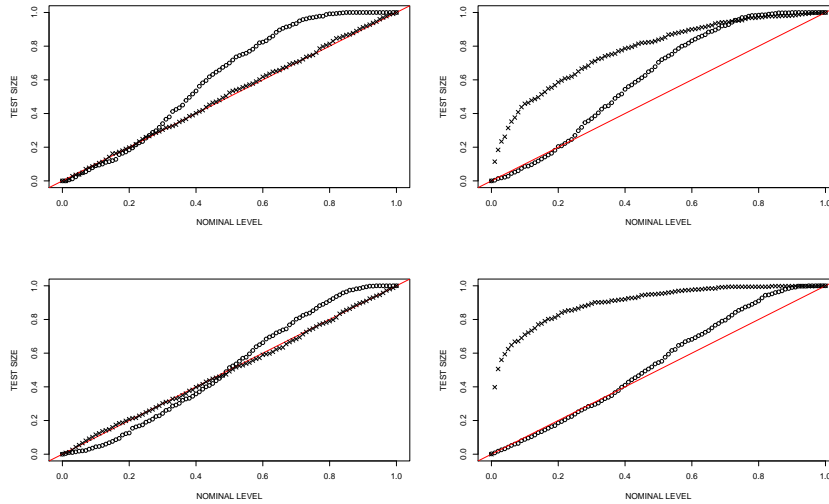

Figure 4.1: Empirical test sizes vs nominal test levels based on 500 experiments. Data are generated based on $\mathcal{M}_{\mathbf{X}}(T_\star)$ with parameters as prescribed in the text. Upper panels: $(m, n) = (20, 250)$. Lower panels: $(m, n) = (20, 500)$. Left panels: Setup 1. Right panels: Setup 2. Open circles: Test based on the statistic $\mathcal{T}$. Crosses: LR test implemented by `factanal`.

## 4.1 Low dimensional setup

We first consider two experimental setups, each with data generated from the one-factor model in (3.1) for both $(m, n) = (20, 250)$ and $(m, n) = (20, 500)$. The model parameters are as follows: (i) Setup 1: all loadings $\beta_p$ and error variances $\sigma_{p,\epsilon}^2$ are taken to be 1. (ii) Setup 2: $\beta_1$ and $\beta_2$ are taken to be 10, while the other loadings are independently generated based on a normal distribution with mean 0 and variance 0.2. The error variances $\sigma_{p,\epsilon}^2$ all equal $1/3$.

For different nominal test levels $\alpha$ in the range $(0, 1)$ that are 0.01 apart, we compare the empirical sizes of our test based on the statistic $\mathcal{T}$ and the likelihood ratio (LR) test implemented by the function `factanal`, using 500 repetitions of experiments. The results are shown in Figure 4.1. The left two panels correspond to Setup 1 and the right two panels correspond to Setup 2, while the upper panels correspond to $(m, n) = (20, 250)$ and lower correspond to $(m, n) = (20, 500)$. While we show the entire range $(0, 1)$ for the x-axis, practical interest is typically in the initial part where the nominal error rate is in say $(0, 0.1)$.

In Setup 1, for both sample sizes, the empirical test sizes of the LR test align almost perfectly with the $45°$ line as one would expect from classical theory. The sizes of our test based on $\mathcal{T}$ also align better with $45°$ line as sample sizes grow. Note that for nominal test levels that are of practical interest, $\mathcal{T}$ also gives conservative test sizes for both sample sizes.

In Setup 2, where parameters are close to being "singular", one can see the true advantage of using $\mathcal{T}$ over the LR test. The empirical test sizes of the LR test with `factanal` do not align well with the $45°$ line as one normally expect from classical theory, whereas the test sizes of our statistic $\mathcal{T}$ lean closer to the $45°$ line as $n$ increases. Particularly the performance of the LR test is problematic since, by rejecting the true model (3.1) all too often, it fails to give even an approximate control on type 1 error. Note that the values of $\beta$ and $\sigma_{p,\epsilon}$ are such that, for the most part, the observed variables $\mathbf{X}$ are rather weakly dependent on each other. If the observations were in fact independent then the likelihood ratio test statistic does not exhibit a chi-square limiting distribution [Drton, 2009, Theorem 6.1]. This highlights the fact that, in addition to avoiding any non-convex optimization of the likelihood function of the factor model, our approach based on the simple estimates from (3.3) is not subject to non-standard limiting behaviors that plague the LR test when the parameter values lean close to singularities of the parameter space [Drton, 2009].

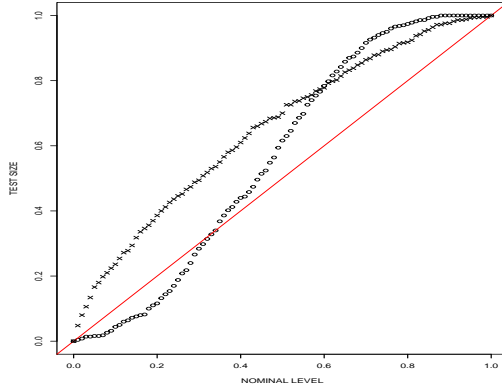

Figure 4.2: Empirical test size vs nominal test levels based on 500 experiments for data generated from $\mathcal{M}_{\mathbf{X}}(T_\star)$ under Setup 1 and $(m, n) = (100, 250)$. Open circles: Test based on $\mathcal{T}$. Crosses: LR test implemented by `factanal`.

## 4.2 Higher dimensional setup

Our last experiment aims to compare the test sizes of the two tests when the number of observed variables $m$ is relatively large compared to $n$. Data are exactly as in Setup 1, except that $(m, n) = (100, 250)$. For such a model with large $m$, the number of tetrads involved in our testing methodology is so large that even after taking the supremum norm one shouldn't expect (3.5) to hold; for example, when $m = 50$, the dimension of $\Theta$ is $2 \cdot \binom{50}{4} = 460600$, and one should be skeptical about the validity of (3.5) when we only have the sample size $n = 250$. To implement our test, we first randomly select 10000 of the $2 \cdot \binom{m}{4}$ tetrads, and proceed with the bootstrapping procedure in (3.8) with $\mathbf{Y}_i$ being estimates for this selected subset of tetrads alone. The choice of 10000 tetrads to be tested is based on the fact that, in the previous experiments with $(m, n) = (20, 250)$, our test gives reasonable empirical test sizes for a practical range of nominal levels when the total number of tetrads being tested, $2 \cdot \binom{20}{4}$, is approximately 10000. Since the subset of tetrads is randomly selected, our test is still expected to approximately control the test sizes at nominal level. The results are reported in Figure 4.2 .

As seen, the test based on $\mathcal{T}$ has the main features seen in the first experiment. In particular, it successfully controls type I error rates for the practical range of $\alpha \in (0, 0.1)$. In contrast, with $m$ increased to 100, the LR test drastically fails to control type I error rate. This is despite the fact that the setup is regular with parameter values that are far from any model singularity. The reason for the failure of the LR test is the fact that the dimension is on the same order as the sample size of 250. The sample size is not large enough for chi-square asymptotics based on fixed dimension $m$ to "kick in".

## 5 Discussion

In this paper we have established a full set of polynomial constraints on the covariance matrix of the observed variables, in the form of both equalities and inequalities, that characterizes a general Gaussian latent tree model whose observed nodes are not confined to be the leaves. Focusing on the special case of a star tree model, we also experimented with a new methodology for testing the equality constraints by forming unbiased estimates of the polynomials involved. In simulation studies, when the number of variables involved is large or the underlying parameters are close to being "singular", our test compares favorably with the likelihood ratio test in terms of test size.

Our results have paved the way for developing a full-fledged algebraic test for a Gaussian latent tree model. Although we have not pursued this generality in the present conference paper, we give a brief discussion here. Of course, to do so one would first need to write an efficient graph algorithm to tease out all the polynomials entailed by Corollary 2.2 for a given latent tree input. Then the current testing methodology can be adopted by forming unbiased estimates of all these polynomials at hand, which also brings to our attention that in Section 3 only the equality constraints in Corollary 2.2$(ii)$ were used to test the single factor model. For illustration, take the 3-degree monomial in Corollary 2.2 (i)(a) as

an example. Like (3.3), one may form a summand $Y_{i,(p,q,r)} = X_{p,i} X_{q,i} X_{p,i+1} X_{r,i+1} X_{q,i+2} X_{r,i+2}$, which is unbiased for $\sigma_{pq}\sigma_{pr}\sigma_{qr}$, and then use $(n-2)^{-1}\sum_{i=1}^{n-2} Y_{i,(p,q,r)}$ as an averaged estimator. To incorporate the constraints in Corollary 2.2 (i) into our test one can first arrange all those inequalities into "less than" conditions, i.e., Corollary 2.2 (i)(a) becomes $-\sigma_{pq}\sigma_{pr}\sigma_{qr} \leq 0$ and the corresponding estimate becomes $-(n-2)^{-1}\sum_{i=1}^{n-2} Y_{i,(p,q,r)}$. Following that, in the definition of the test statistic $\mathcal{T}$, one can take a maximum over all the unbiased estimates for the "less than" versions of the polynomials in Corollary 2.2$(i)$, in addition to the absolute values of the estimates for the polynomials in Corollary 2.2$(ii)$. The resulting test statistic shall also reject the model $\mathcal{M}(T_\star)$ when its value is too large. While critical values can still be calibrated with multiplier bootstrap, additional techniques such as inequality selection can be incorporated to contain the power loss as a result of testing the inequalities; see Chernozhukov et al. [2013b] for more details.

Another challenge is the determination of the batch size $B$ in (3.6). In our simulation studies of Section 4 we took $B = 3$ since we believe that a batch size of 3 should be enough to capture dependence among the 1-dependent summands. Batch size determination has been widely studied in the time series literature for low dimensional problems [Bühlmann, 2002, Hall et al., 1995, Lahiri, 2003]. To the best of our knowledge, in high dimensions this is still a widely open problem. Theoretical research on this is far beyond the scope of our current work.

### Acknowledgments

Part of this work was undertaken while Dennis Leung was a postdoc at the Chinese University of Hong Kong, and he would like to thank Professor Qi-Man Shao for some helpful discussions during that time.

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
