[Supplementary Material]

# Supplement for: Algebraic tests of general Gaussian latent tree models

**Dennis Leung**
Department of Data Sciences and Operations
University of Southern California
dmhleung@uw.edu

**Mathias Drton**
Department of Statistics, University of Washington &
Department of Mathematical Sciences, University of Copenhagen
md5@uw.edu

In this supplement we furnish proofs for the main text of "Algebraic tests of general Gaussian latent tree models".

## 6   Proof of Corollary 2.2

We only sketch the proof here since it is exactly analogous to that of Theorem 3 in Shiers et al. [2016]. First, consider the special case where all the entries of $\Sigma$, and hence the the Pearson correlations $\rho_{pq}$, $1 \leq p \neq q \leq m$, are strictly positive. In this case condition $(i)(a)$ is redundant. Via the isomorphsim

$$\delta_{pq} = -\log \rho_{pq},$$

between the parametrizations in (2.1) and all $T$-induced pseudometrics, the discussion preceding our corollary readily translates (2.2) into $(i)(c), (ii)(b), (ii)(c)$ and (2.3) into $(ii)(a)$, whereas the triangular inequality property of pseudometrics is translated into $(i)(b)$ for triples $\{p, q, r\}$ that are not in $\mathcal{L}$. The general case of $\Sigma$ with nonzero but not necessarily positive entries is then addressed by incorporating condition $(i)(a)$.

## 7   Proof of Lemma 2.1

To prove the lemma, we first collect all the required graphical notions borrowed from Semple and Steel [2003]. We attempted to make this proof as self-contained as possible, but the readers are encouraged to read Semple and Steel [2003] for more background on mathematical phylogenetics.

Suppose we are given a tree $T = (V, E)$. If $\tilde{V}$ is a subset of $V$, $T(\tilde{V})$ denotes the minimal subtree of $T$ that contains all the nodes in $\tilde{V}$. If $e \in E$, $T \backslash e$ is the graph obtained by removing $e$, and $T/e$ is the tree obtained from $T$ by identifying the ends of $e$ and then deleting $e$. In particular, if $v \in V$ is a node of degree two and $e$ is an edge incident with $v$, $T/e$ is said to be obtained from $T$ by *suppressing* $v$. If $v_1, v_2 \in V$, $ph_T(v_1, v_2)$ is the set of edges on the unique path connecting $v_1$ and $v_2$.

We will also need the notion of an **X**-tree. An **X**-tree, or semi-labeled tree on a set **X**, is an ordered pair $\mathcal{T} = (T, \phi)$, where $T$ is a tree with node set $V$ and $\phi : \mathbf{X} \to V$ is a (labeling) map with the property that, for each $v \in V$ *of degree at most two*, $v \in \phi(\mathbf{X})$. Note that $\phi$ is not necessarily injective. Moreover, if $\mathbf{X}'$ is a subset of **X**, $T|\mathbf{X}'$ is the tree obtained from $T(\phi(\mathbf{X}'))$ by suppressing all the nodes of degree two that are not in $\phi(\mathbf{X}')$. We then define the *restriction of $\mathcal{T}$ to $\mathbf{X}'$*, denoted $\mathcal{T}|\mathbf{X}'$, to be the $\mathbf{X}'$-tree $(T|\mathbf{X}', \phi|\mathbf{X}')$.

Finally, we introduce the notion of **X**-*split*. For a set **X**, an **X**-split is a partition of **X** into two non-empty sets. We denote the **X**-split whose blocks are $A$ and $B$ by $A|B$ where the order of $A$ and

$B$ in the notation doesn't matter. Now suppose $\mathcal{T} = (T, \phi)$ is an **X**-tree with an edge set $E$. For each $e \in E$, $T \backslash e$ must consist of two components $V_1^e$ and $V_2^e$ which induce an **X**-split $\phi^{-1}(V_1^e) | \phi^{-1}(V_2^e)$. We then define $\Sigma(\mathcal{T}) := \{\phi^{-1}(V_1^e) | \phi^{-1}(V_2^e) : e \in E\}$ as the collection of all **X**-splits induced by $\mathcal{T}$.

*Important remark*: In all the definitions above, **X** is not specified as a subset of the node set $V$ for a given tree. Nonetheless, when we have a tree $T = (V, E)$ with a subset of observed nodes $\mathbf{X} \subset V$ as in the main text, we will slightly abuse the notations by identifying $T$ with the **X**-tree whose labeling map is simply the identity function. Moreover, if $\mathbf{X}' \subset \mathbf{X}$, we will also identify $T | \mathbf{X}'$ with the **X**'-tree that is the restriction of $T$ (as an **X**-tree) to **X**'.

Now we begin to prove Lemma 2.1. The "only if" part of the theorem is trivial and we will only prove the "if" part of the statement.

We recall that $\mathbf{X} = \{X_1, \ldots, X_m\}$. Let $\delta$ be a pseudo-metric on **X** satisfying the two conditions (2.2) and (2.3) in display. For any four distinct points $p, q, r, s \in [m]$, given the tree structure of $T$ it must be true that $ph_T(\pi_p, \pi_q) \cap ph_T(\pi_r, \pi_s) = \emptyset$ for some permutation $\pi$ of $p, q, r, s$. By (2.2), together with the fact that $\delta$ is a pseudo-metric, $\delta$ is in fact a *tree metric* (Semple and Steel [2003, Theorem 7.2.6]), i.e., there exists an **X**-tree $\tilde{\mathcal{T}} = (\tilde{T}, \tilde{\phi})$ for a tree $\tilde{T} = (\tilde{V}, \tilde{E})$ and a labeling map $\tilde{\phi} : \mathbf{X} \to \tilde{V}$, as well as a *strictly positive* weighting function $\tilde{w} : \tilde{E} \longrightarrow \mathbb{R}_{>0}$ such that

$$\delta_{pq} = \begin{cases} \sum_{\tilde{e} \in ph_{\tilde{T}}(\tilde{\phi}(X_p), \tilde{\phi}(X_q))} \tilde{w}(\tilde{e}) & \text{if } \tilde{\phi}(X_p) \neq \tilde{\phi}(X_q) \\ 0 & : \tilde{\phi}(X_p) = \tilde{\phi}(X_q) \end{cases}$$

for all $p, q \in [m]$. By Theorem 6.3.5$(i)$ and Lemma 7.1.4 in Semple and Steel [2003], to show that $\delta$ can be induced from $T$ it suffices to show that for any $\mathbf{X}' \subset \mathbf{X}$ of size at most 4, the two restricted **X**'-trees $\tilde{\mathcal{T}} | \mathbf{X}'$ and $T | \mathbf{X}'$ are such $\Sigma(\tilde{\mathcal{T}} | \mathbf{X}') \subset \Sigma(T | \mathbf{X}')$. Note that this is trivial for $|\mathbf{X}'| = 1$ and $|\mathbf{X}'| = 2$. For $3 \leq |\mathbf{X}'| \leq 4$, we first note that

$$\{X_p\} | \mathbf{X}' \backslash \{X_p\} \in \Sigma(\tilde{\mathcal{T}} | \mathbf{X}') \text{ if and only if } \delta_{pq} + \delta_{pr} - \delta_{qr} > 0 \text{ for all } X_q, X_r \in \mathbf{X}' \backslash \{X_p\} \quad (7.1)$$

and

$$\{X_p, X_q\} | \{X_r, X_s\} \in \Sigma(\tilde{\mathcal{T}} | \mathbf{X}') \text{ if and only if}$$
$$\delta_{pr} + \delta_{qs} - \delta_{pq} - \delta_{rs} > 0 \text{ ( and } \delta_{ps} + \delta_{qr} - \delta_{pq} - \delta_{rs} > 0). \quad (7.2)$$

These characterization for the elements in $\Sigma(\tilde{\mathcal{T}} | \mathbf{X}')$ can be easily checked; also see Semple and Steel [2003, p.148] where these characterizations are stated. To finish the proof it remains to show that, when $3 \leq |\mathbf{X}'| \leq 4$, any **X**'-split $\{X_p\} | \mathbf{X}' \backslash \{X_p\}$ as in (7.1) or any **X**'-split $\{X_p, X_q\} | \{X_r, X_s\}$ as in (7.2) must also be an element of $\Sigma(T | \mathbf{X}')$.

First, towards a contradiction, suppose there exists an **X**'-split $\{X_p\} | \mathbf{X}' \backslash \{X_p\}$ that is an element of $\Sigma(\tilde{\mathcal{T}} | \mathbf{X}')$ but not an element of $\Sigma(T | \mathbf{X}')$. Since $\{X_p\} | \mathbf{X}' \backslash \{X_p\}$ is not an element of $\Sigma(T | \mathbf{X}')$, by considering $T | \mathbf{X}'$ as a tree it must be the case that the node $X_p$ has degree at least two. Then, by condition (2.3), there must exist two distinct $X_q$ and $X_r$ in the set $\mathbf{X}' \backslash \{X_p\}$ such that $\delta_{pq} + \delta_{pr} = \delta_{qr}$. But this reaches a contradiction since by (7.1) $\delta_{pq} + \delta_{pr} - \delta_{qr} > 0$ as $\{X_p\} | \mathbf{X}' \backslash \{X_p\} \in \Sigma(\tilde{\mathcal{T}} | \mathbf{X}')$.

Similarly, suppose $\{X_p, X_q\} | \{X_r, X_s\}$ is an element of $\Sigma(\tilde{\mathcal{T}} | \mathbf{X}')$ but not an element of $\Sigma(T | \mathbf{X}')$. Since $\{X_p, X_q\} | \{X_r, X_s\} \in \Sigma(\tilde{\mathcal{T}} | \mathbf{X}')$, the two strict inequalities in (7.2) must be true. On the other hand, if $T | \mathbf{X}'$ has any of the configurations in Figure 2.2$(a) - (c)$, since $\{X_p, X_q\} | \{X_r, X_s\} \notin \Sigma(T | \mathbf{X}')$ it must be true that $ph_T(p, q) \cap ph_T(r, s) \neq \emptyset$, in which case it must lead to either $ph_T(p, r) \cap ph_T(q, s) = \emptyset$ or $ph_T(p, s) \cap ph_T(q, r) = \emptyset$, contradicting one of the inequalities in (7.2) by condition (2.2). If $T | \mathbf{X}'$ has the configuration in Figure 2.2$(d)$ or $(e)$, then it must be the case that both $ph_T(p, r) \cap ph_T(q, s)$ and $ph_T(p, s) \cap ph_T(q, r)$ are empty sets, which also contradict both inequalities in (7.2) by condition (2.2).