[Reviews · NeurIPS 2018]

Reviewer 1



Paper Summary: The paper presents a technique for testing whether a given set of samples are drawn from a postulated Gaussian latent tree model or a saturated Gaussian graphical model. The paper first characterizes a set of necessary and sufficient constraints that any covariance matrix of a Gaussian latent tree model should satisfy. It then uses these constraints to come up with a test statistic. The paper extends past work on testing for Gaussian latent tree models to settings where the observed variables are allowed to have degree up to 2. The test statistic presented in the paper is based on gaussian approximation for maxima of high dimensional sums. Simulations suggest that the test statistic can potentially work in high dimensional settings. Comments: 1) Section 3 is mostly focused on star tree models. The general setting is covered only briefly in the last section. I would have preferred more details about the general setting. If the inequality constraints are considered, will the resulting test statistic still have the distribution in lines 201-202? What will be the exact test when inequalities are considered? 2) [Equation 3.5] For what kind of dependencies among Y_i's does the equation hold? Will the dependencies among Y_i's defined in (3.3) satisfy the required conditions for 3.5 to hold? 3) [High dimensions:] The proposed approach needs to estimate a covariance matrix, which can be rank deficient and very inaccurate when the number of tetrads is larger than the number of samples. As a result the approximation in lines 201-202 could be inaccurate. Given this, I'm surprised that the proposed estimator performed so well in section 4.1. Can this behavior be explained? 4) The paper is mostly well written, however there are some sections which need more details. For example, in line 160, it is not clear what is the "gradient" of tetrad. Lines 159-162 will need more explanation. In line 91, what is a minimal subtree? Also, the proof of Lemma 2.1 will need some more details (especially lines 21-25). In line 21, what is T'|S \leq T|S?

Reviewer 2



Summary: The authors consider Gaussian latent tree models for which they derive algebraic polynomial constraints. These constraints extend the results of Shiers et al. [2016] and allow for the observed variables to correspond to nodes with degree larger than one (hence observed variables are restricted to being leaf nodes). Based on the polynomial constraints, the authors construct a goodness-of-fit test for the restricted class of star tree models. This test is based on a bootstrap approximation of the asymptotic distribution, in a way that makes this test feasible for practically relevant data sizes. Finally, the proposed test is empirically assessed compared to a LRT for the one factor factor model (which corresponds to the star tree structure). Detailed comments: As this is not my field of expertise, I will not judge the novelty of this paper. However, overall the paper is well written and nicely structured and to the best of my knowledge the results of the paper seem correct. There are, nevertheless, some questions/concerns I had while reading: (1) The first part of the paper (Section 2) extends the results of Shiers et al. [2016] to tree models in which the observed variables do not need to be leaf nodes. While this extension seems interesting it is later not used in the construction of goodness-of-fit tests in Section 3. If the construction of goodness-of-fit tests to more general tree models is too hard, why did the authors introduce the more complicated extension to the tree models in the first place? (2) The results given in Figure 4.1 were rather surprising to me. If the proposed procedure does indeed approximate the exact asymptotic distribution, why does the test become more conservative when the sample size is increased from 250 to 500 (top vs bottom left) for low levels? Also for the LRT why does the method become worse in reaching nominal level for larger sample sizes? Again this test should be asymptotically consistent. (3) In the experiment with Setup 2 the results are quite promising for the proposed procedure. However, to my understanding this is a setting that is close to a model violation for the LRT, therefore it would also be interesting to see how the proposed procedure would behave in terms of a model violation of its own assumptions. For example, if one uses an underlying factor model with two unknown factors.

Reviewer 3



# After authors response I thank the authors for their detailed answers and explanations, that corrected some of the misconceptions I had. I acknowledge that the application of the methodology of section 3 to the general framework of section 2 would require some further developments, that might be outside of the scope of this conference paper. Hence, the problem of the discrepancy between the two sections remains for this submission, although both sections are interesting in themselves. # Summary This well written and nicely presented work is mainly interested in the algebraic characterization and statistical test of a latent Gaussian tree structure. This problem has received some attention over the last few years, and can be related to the work of Shiers et al. 2016 (or Zwiernik et al. 2016 although with a different focus). Section 2 focuses on the mathematical derivation of polynomial constraints that characterize variance matrices induced by Gaussian latent tree models. It follows closely the derivations of Shiers et al. 2016, extending them to the case where observations can be anywhere on the tree, including on inner nodes, and not just its leaf nodes. The results are presented in an intuitive way, and the proofs are given in the appendices. In Section 3, a test statistic is derived, in the simplified case of a star-tree. A link is drawn between this star-tree case and the tetrads problem, as studied in the factor analysis literature. The derivation of the new statistics and its approximate distribution relies on recent work by Chernozhukov et al. 2013 and Zhang and Wu 2017. The statistic is justified by heuristic argument, but the proofs are not given. In Section 4, some numerical experiments are conducted to study the empirical levels of the proposed test. It is compared with a standard maximum-likelihood test as traditionally used in factor analysis (function factanal in base R). # Main comments My main concern is about the coherence and unity of the ideas presented. In Section 2, the authors describe with much mathematical details a new framework allowing observations to be at inner nodes of a tree. However, in section 3 and following, only the case of a star tree is considered, with observations only at leaf nodes. Maybe I missed something and I would be happy to be corrected, but from my understanding all the developments in Section 3 could have been made with results already exposed in Shier et al. 2016, their main inspiration for Section 2. Hence, as it stands, section 2 seems somewhat decorrelated from Section 3, and, although the results presented in Section 2 are theoretically interesting in themselves, it is not clear what they bring in the progression of the article. As a minor note related to the same point, only one reference is given to justify the need of the theoretical developments of section 2 (l.34). Adding more examples could help the reader understanding the pertinence of extending existing results to allow for inner-node measurements. That being said, I tend to think that both parts of the paper (section 2 and sections 3-4) are valuable advances to the field, and have the potential to be the starting point for further developments. I found the results of section 2 and very well presented. The background, as well as the novelty of the results, are both clearly exposed. Section 3 makes use of novel statistical techniques to build a new test (heuristically justified) that might have the potential to be used, beyond the star tree, to the more general structures presented in section 2. # Other issues ## Section 2 While I spent a reasonable amount of time to try to understand the results in themselves, I did not review the proofs in details. I found that the new notions and differences with previous characterisations where well explained, and the intuitions were easy to follow. Other comments: * Why are the observations limited to nodes with degree less or equal than 2 ? Is this a constrain inherited from applications, or would the developments of the paper be wrong otherwise ? From Fig. 2-e, it does not prevent sub-trees to have observed nodes of degree 3. * I did not see why a pseudo-distance is needed, while a distance seems to be used in Shiers et al 2016 ? * Cor. 2.2 is linked with Th. 3 in Shiers et al, the same way Lemma 2.1 is linked with Cor 1. Although the links and differences between Lemma 2.1 and Cor 1 are clearly mentioned, the relationships between Cor 2.2 and Th 3 remain allusive. An intuitive description of these differences might help the reader understand the new constraints implied by the new problem. * Fig 2.1 and 2.2 are very helpful. Could it be useful to add as legends {p, q, r, s} such that {p, q}|{r, s} on Fig 2.2 a-c ? ## Sections 3 - 4 While section 2 seems theoretically sound, the results in section 3 are only justified heuristically, and heavily rely on cited references. Since I'm not very familiar with these techniques, I only have a few remarks about the form: * l.141-142: should it be \epsilon_1, …, \epsilon_m and X_1, …, X_m (m instead of p) ? * The authors compare their test statistics with the one in Shier et al 2016 at the end of Section 3, but then do not include this method in the numerical experiments of Section 4. Are there arguments, other than computational time (which could be a legitimate concern) to exclude it ? * Fig. 4.1: Maybe a legend on the rows/columns of the figure could help. Also, since the interval (0, 0.1) is particularly significant, maybe a visual aid, such as a vertical line, could help. * l.255-256: the statement "The sizes of our test based on T also align better with 45° line as the sample sizes grow." is not clear to me in the context of Setup 1 (which is discussed in this paragraph). In particular, it does not seem to be the case in the interval of interest (0, 0.1). * Could a quantitative score, such as the area between the empirical curves and the 45° line, be useful to make more objective statements about the performances of the methods ? * It might be interesting to get an idea of the computational times involved by the application of these tests. ## Reproducibility The test statistics as well as the numerical experiments are well described, so that it should be possible for a researcher willing to reproduce these analyses to write the necessary code. However, no package, or R script, is provided to make this exercise easier.